# Research on the Auto-Exposure Method of an Aerial TDI Camera Based on Scene Prediction

**Jingtao Huang** [1,2], **Jiwei Liu** [1,2], **Xiaodong Wang** [1,2] and **Xu Jiang** [3,*]

1   Changchun Institute of Optics, Fine Mechanics and Physics, Chinese Academy of Sciences,
    Changchun 130033, China; huangjt@ciomp.ac.cn (J.H.); wangxd@ciomp.ac.cn (X.W.)
2   Key Laboratory of On-Orbit Manufacturing and Integration for Space Optics System,
    Chinese Academy of Sciences, Changchun 130033, China
3   Changchun Automobile Industry Institute, College of Electrical Engineering, Changchun 130013, China
*   Correspondence: 3511643@163.com

**Abstract:** Aerial TDI cameras can obtain images with a high sensitivity, high resolution, and wide dynamic range under low-illumination conditions. Under the condition of low illumination, exposure is an important parameter that affects the image quality of the camera. Auto-exposure aims to obtain the information of target scene in advance and uses it as the basis for determining exposure parameters, which can avoid the loss of image information caused by overexposure or underexposure. The auto-exposure of TDI CCD is usually difficult, as the shooting mode of TDI CCD is a push and sweep mode, which can only take one image of the target scene, and the output image is not repetitive, so it is difficult to obtain an image of the target scene in advance. At present, the common method is to add an additional sensor to collect the feature information of the target scene in advance; however, this increases the complexity of the system. Therefore, this paper proposes that the camera uses TDI CCD to collect the first N rows of data of the target scene in advance as the basis, and then adjusts and determines the exposure parameters with the median gray value as the target; thus, without adding additional sensors, the auto-exposure of TDI CCD can be realized. To evaluate the effect of auto-exposure, three methods of image power spectrum variance, image histogram, and image information entropy are used. The test results show that, after auto-exposure, the variance in the image power spectrum increased by 0.4362, the entropy of image information increased by 1.7064, and the distribution of the image histogram was more uniform than that before auto-exposure. This shows that the effect of auto-exposure is good, and it has better scene adaptation, allowing for it to meet the requirements of auto-exposure imaging under low-illumination conditions in aerial TDI cameras.

**Keywords:** TDI CCD; auto-exposure; aerial camera; scene prediction




## 1. Introduction

Since the first automatic film aerial camera K-1 was invented in 1921, aerial cameras have become the key equipment of remote sensing imaging and photogrammetry [1,2]. Early aerial cameras were frame-type due to the slow speed of their center shutter, which would result in large image motion fuzziness when the aircraft was flying at a high velocity–height ratio. This led to their gradual replacement by push and sweep cameras [3,4]. In early push and sweep aerial cameras, there was no shutter and only a slit was placed in the focal plane. The film quickly passed through the slit to form a continuous image [5,6]. Therefore, in order to compensate for the forward image motion produced in the exposure process, the motion speed of the film had to be synchronized with the flight speed, which increased the complexity of the system [7]. With the continuous progress in the manufacturing level of remote sensing cameras and the development of semiconductor technology, remote sensing cameras developed from the era of film to the era of electronic image sensors, which caused the electronic means of forward image motion compensation

technology to begin to rise; this is known as Time Delayed and Integration (TDI) [8,9]. At present, this technology, whose basic principle is to expose the same scene multiple times and add up the exposed images, has been widely used in push and sweep aerial cameras [10–12].

The shooting distance of aerial remote sensing cameras is usually relatively long, and the radiation energy of ground scenes will be attenuated after passing through the atmosphere and lens [13]. In order to obtain more radiation energy under the condition of low illumination, the direct method increases the exposure time, but a longer exposure time increases the fuzzy quality of image motion [14,15]. Therefore, by using TDI technology, more radiation energy can be obtained without increasing the exposure time. In aerial TDI cameras, TDI CCD, which is a time-delayed integral charge-coupled device, is usually used as the sensor [16], as it can obtain high-sensitivity, high-resolution, and large-dynamic-range images using TDI technology under the condition of low illumination [17].

With the development and progression of science and technology, semiconductor technology is becoming more and more precise, and the difficulties in manufacturing aerial remote sensing cameras are increasing; therefore, improving image quality is one of the problems to which researchers have been committed [18]. There are many factors affecting the image quality of aerial remote sensing cameras [19], one of which is exposure; this is the main reason for the decline in image quality under the condition of low illumination, as this determines the degree of light energy absorption of the cameras, changes the overall gray value of the image, and affects the visual effect and information in the image. Auto-exposure aims to obtain the information of the target scene in advance, and uses this as the basis for determining parameter exposure, which can avoid the loss of image information caused by overexposure or underexposure. The auto-exposure of TDI CCD is usually difficult, as the shooting mode of TDI CCD is push and sweep, which can only take one image of the target scene, and the output image is not repetitive, so it is difficult to obtain the image of the target scene in advance. At present, the common method is to add an additional sensor to collect the feature information of the target scene in advance; however, this increases the complexity of the system [20,21].

In order to achieve auto-exposure in aerial TDI cameras, this paper studies the imaging link model, and proposes that the camera uses TDI CCD to collect the first N rows of data of the target scene in advance as the basis, and then adjusts and determines the exposure parameters with the median gray value as the target; thus, in the case where there are no additional sensors, the auto-exposure of TDI CCD can be realized.

## 2. Auto-Exposure Method

The purpose of auto-exposure is to change the gray value of the image; therefore, by adjusting the exposure parameters, the gray value can approach the target gray value, and it is generally considered that the average gray value of the image obtained by the aerial TDI camera is more conducive to image processing when it is half full. The Auto-exposure of TDI CCD is realized by capturing images of the target scene in advance, and collecting the average gray value of the front N rows, which is used for comparison with the gray value of the target; therefore, setting the gray value of the image to approach the target value by adjusting the exposure parameters.

The auto-exposure of an aerial TDI camera is a complex process. To achieve auto-exposure, an imaging link model of an aerial TDI camera is established to analyze the imaging process, as shown in Figure 1.

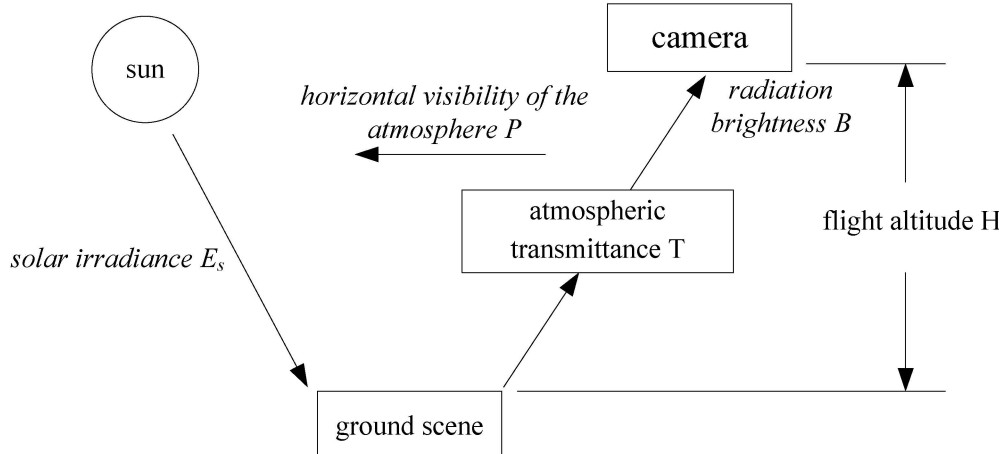

**Figure 1.** Imaging link model of aerial TDI camera.

In the process of earth remote sensing imaging, sunlight is the main source of illumination. The ground scene can obtain the radiation brightness by reflecting the solar illumination; however, the radiation brightness will be weakened to a certain extent through the absorption and scattering of the atmosphere, and finally be acquired by the camera lens.

As can be seen from Figure 1, the solar irradiance described by $E_S$ is measured on the ground by an illuminometer, whose unit is lx, and the relation between lx and the Luminous flux (lm) is:

$$1\text{lx} = 1\text{lm}/\text{m}^2 = 1/683V(\lambda)\text{W·m}^{-2}, \tag{1}$$

where $V(\lambda)$ is a normalized visual function [22], and, according to CIE 1931 [23], its value is 0.28 in the visible wavelength range $\lambda = 390 \sim 780$ nm.

The relation between irradiance $E$ of the target and the solar irradiance $E_S$ is:

$$E = \frac{E_S}{683V(\lambda)}. \tag{2}$$

In Equation (2), the unit of $E$ is W/m$^2$, and, according to the relation between the radiation irradiance and the radiation luminance of Lambert body, the radiation luminance $L$ of the target is obtained by:

$$L = \frac{E \cdot \rho_A}{\pi} = \frac{E_S \cdot \rho_A}{683V(\lambda)\pi}, \tag{3}$$

where $\rho_A$ refers to the average reflectivity of the ground scene.

The brightness $B$ has the relation between $L$ and the atmospheric transmittance, that is:

$$B = L \cdot T = \frac{E_S \cdot \rho_A \cdot T}{683V(\lambda)\pi}, \tag{4}$$

where $T$ donates the atmospheric transmittance, which is the ratio of the outgoing light to the incident light, and it is given by:

$$T = \frac{I(O)}{I(I)}, \tag{5}$$

where $I(O)$ is the outgoing light intensity and $I(I)$ refers to the incident light intensity.

According to the Beer–Lambert law, the relation between the intensity of the outgoing light and the intensity of the incident light is:

$$I(O) = I(I) \times e^{-KM},$$ (6)

where $K$ donates the extinction coefficient and $M$ represents the optical path length.

Thus, Equation (5) can be rewritten as:

$$T = \frac{I(O)}{I(I)} = e^{-KM}.$$ (7)

A logarithm of e-based is taken on both sides of Equation (7), that is $\ln T = -KM$, and it can be obtained $K = -\frac{\ln T}{M}$.

The horizontal visibility, described by $P$, has the relation with $K$:

$$P = \frac{1}{K} \ln(\frac{1}{\varepsilon}),$$ (8)

where $\varepsilon$ refers to the minimum luminance contrast that the human eye can distinguish from a sufficiently large object at a distance, which is called the contrast perception threshold [24], and is usually 0.02.

If $\varepsilon = 0.02$ and $K = -\frac{\ln T}{M}$ are substituted into Equation (8), it can be expressed as $P = \frac{1}{K} \ln(\frac{1}{\varepsilon}) = -\frac{1}{K} \ln(0.02) = \frac{3.91}{K}$, therefore, $P$ is obtained:

$$P = -\frac{3.91M}{\ln T}.$$ (9)

Take e exponents on both sides of Equation (9), therefore, $T$ is obtained:

$$T = e^{-3.91\frac{M}{P}},$$ (10)

where $P$ refers to the horizontal visibility of the atmosphere, and it is taken vertically down to the ground during a flight, so the optical path length $M$ is equal to the flying altitude $H$; therefore, Equation (10) can be rewritten as $T = e^{-3.91\frac{H}{P}}$.

If Equation (10) is substituted into Equation (4), it can be expressed as:

$$B = \frac{E_S \cdot \rho_A \cdot T}{683V(\lambda)\pi} = e^{-3.91\frac{H}{P}} \cdot \frac{E_S \cdot \rho_A}{683V(\lambda)\pi}.$$ (11)

Since the radiation brightness $B$ has been obtained, we can establish the camera's processing flow of the light signal. This is the process of converting the light signal of the ground scene into the electrical signal of digital image, as shown in Figure 2.

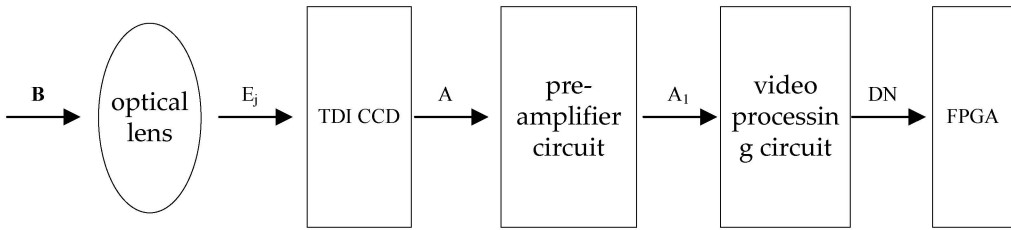

**Figure 2.** Camera's processing flow of the light signal.

The optical lens is the entrance of the camera to receive the optical signal, which converges the optical signal of the target scene onto the TDI CCD sensor. The optical signal at the entrance of the lens is the radiation brightness $B$ at the entrance pupil of the camera,

and, after passing through the optical system, it becomes the irradiance $E_j$ of the sensor. The relation between $E_j$ and $B$ is:

$$E_j = \frac{\pi}{4} \times \tau \times \frac{1}{F^2} \times B,$$ (12)

where $\tau$ and $F$ represent the average transmittance and relative aperture of the optical system.

The TDI CCD converts the optical signal into an analog electrical signal, denoted as $A$, which is given by:

$$A = E_j \times t \times \frac{R}{96} \times N,$$ (13)

where $t$ refers to the integral time, $R$ donates the responsivity of TDI CCD, whose unit is V/J/M$^2$, and $N$ represents the integral stages.

The analog signal output by the TDI CCD can be amplified and the noise suppressed by the pre-amplifier circuit. The amplified analog signal is denoted as $A_1$, and its expression is:

$$A_1 = A \times g,$$ (14)

where $g$ is the magnification of the pre-amplifier circuit, and, in this design, its value is 1, thus $A_1 = A$.

The amplified analog signal, after going through the process of A/D conversion and correlation double sampling by the video-processing circuit, will become a digital signal. Therefore, the image gray value $DN_{CCD}$ is given by:

$$DN_{CCD} = A_1 \times \frac{1}{G},$$ (15)

where $G$ refers to the electronic quantization gain, whose unit is DN/V, and, in this design, the quantization bit of the video processor is 12 bit, thus $G = 2048$ DN/V.

Based on the above analysis, the imaging mathematical model of the aerial TDI camera can be obtained as follows:

$$DN_{CCD} = \frac{\pi}{4} \times \tau \times \frac{1}{F^2} \times B \times t \times \frac{R}{96} \times N \times \frac{1}{G}.$$ (16)

In Equation (16), the average transmittance of the optical system $\tau$, the relative aperture of the optical system $F$, the responsivity of the TDI CCD sensor $R$, and the electronic quantization gain $G$ are fixed values after the system design is completed, and the radiation brightness $B$ can be calculated by Equation (11). The integral time $t$ needs to be strictly matched with the velocity–height ratio in order to achieve image motion matching; generally, the flight speed and height of the aircraft during a flight can be measured by the position and orientation unit, therefore, the velocity–height ratio is basically constant, thus, the integral time $t$ is a fixed value. So, the direct method to change the gray value of the image is to change the integral stages, therefore, the auto-exposure can be realized by changing the integral stages $N$.

Finally, Equation (16) can be rewritten as:

$$\begin{aligned} DN_{CCD} &= \frac{\pi}{4} \times \tau \times \frac{1}{F^2} \times B \times t \times \frac{R}{96} \times N \\ &= \frac{b \times N}{f \times \eta} \times \frac{\pi}{4} \times \frac{1}{F^2} \times \tau \times e^{-3.91 \times \frac{H}{P}} \times \frac{E_S \times \rho_A}{683 \times V(\lambda) \times \pi} \times \frac{R}{96}, \end{aligned}$$ (17)

and, according to Equation (17), by changing the integral stages $N$, the gray level of the image can approach the target value, so as to realize the auto-exposure.

In this paper, the imaging link model of an aerial TDI camera is studied and the radiation brightness $B$ is obtained; therefore, by establishing the camera's processing flow of the light signal, we obtain the mathematical model of the camera, from which we can know which parameters to adjust to achieve the auto-exposure. Specifically, by collecting

the first 10 rows of data in advance and calculating its average gray value, which is used as the initial value for comparison with the median gray value, and then by changing the integral stages $N$ according to Equation (17), the average gray value of the image can approach the target value. In this way, a properly exposed image can be obtained.

## 3. Auto-Exposure Effect Evaluation

Three methods of image power spectrum variance, image histogram, and image information entropy are used to evaluate the auto-exposure effect. The image power spectrum can represent the information content of the image, and the auto-exposure effect can be evaluated by comparing the image power spectrum variance before and after auto-exposure. The greater the image power spectrum variance is, the more information the image contains and the better the auto-exposure effect will be. The image histogram can directly reflect the brightness distribution and overall exposure of the image; when the histogram is evenly distributed, it means the auto-exposure effect is good. The image information entropy is used to characterize the clarity of the image, and the more evenly distributed on each gray value the gray histogram is, the clearer the image is, and the greater the information entropy of the image is; therefore, this means the auto-exposure is effective.

### 3.1. Image Power Spectrum Variance

The power spectrum reflects the characteristics of the image in the frequency domain, and it is closely related to the information content of the image. Fourier transform is used to transform the signal from the spatial domain to the frequency domain, which is defined as follows:

$$F(u,v) = \sum_{x=0}^{M-1}\sum_{y=0}^{N-1} f(x,y)exp[-2\pi j(\frac{ux}{M} + \frac{vy}{N})],\tag{18}$$

where $F(u,v)$ is the two-dimensional discrete Fourier transform of the image, $f(x,y)$ is an image of size $M$ by $N$, $u$ and $v$ are the frequency components, and Equation (18) must be evaluated in the range $u = 0,1,2,\cdots,M-1$ and $v = 0,1,2,\cdots,N-1$. Two-dimensional discrete Fourier transforms are usually a complex function, which can be expressed in polar coordinates as:

$$F(u,v) = |F(u,v)|e^{j\phi(u,v)},\tag{19}$$

where the amplitude $|F(u,v)|$ is called the Fourier spectrum or spectrum.

The power spectrum is the square of the spectrum, reflecting the energy size of the image in each spatial frequency component, and it is defined as $P(u,v) = |F(u,v)|^2$.

As shown in Figure 3, two clear and fuzzy images are used, respectively, for comparison, Figure 3(a1,b1) are the clear and fuzzy images, Figure 3(a2,b2) are the corresponding power spectra, and Figure 3(a3,b3) are the corresponding three-dimensional power spectra. As can be seen from the figure, Figure 3(a1) is a clear image and there are more bright spots in the Fourier power spectrum, while Figure 3(b1) is a fuzzy image and there are fewer bright spots in the Fourier power spectrum. Therefore, the clarity of the image determines the complexity of its power spectrum; in other words, the higher degree of dispersion of the image power spectrum, the clearer the image is.

In the field of digital image processing, the dispersion of the image is usually represented by variance, and the variance in the image power spectrum is represented by $\delta$,

$$\delta = \frac{1}{M \times N}\sum_{u=0}^{M-1}\sum_{v=0}^{N-1}\left[P(u,v) - \overline{P(u,v)}\right]^2,\tag{20}$$

where $\overline{P(u,v)}$ refers to the average of the image power spectrum, and $M$ and $N$ are the number of rows and columns of the image, respectively.

In order to verify the relationship between the variance in the image power spectrum and the clarity of the image, the lens blur processing of different degrees is carried out on the

clear image in Figure 3(a1). The blur radius is r and the value range of r is [0, 1, 2. . . . . . 10]. The larger the value of r is, the more serious degree of the image blur will be.

As can be seen from Figure 4, the variance in the image power spectrum gradually decreases with an increase in the blur radius. Therefore, it can be concluded that, the clearer the image is, the greater the image power spectrum variance is and the more frequency information the image power spectrum contains.

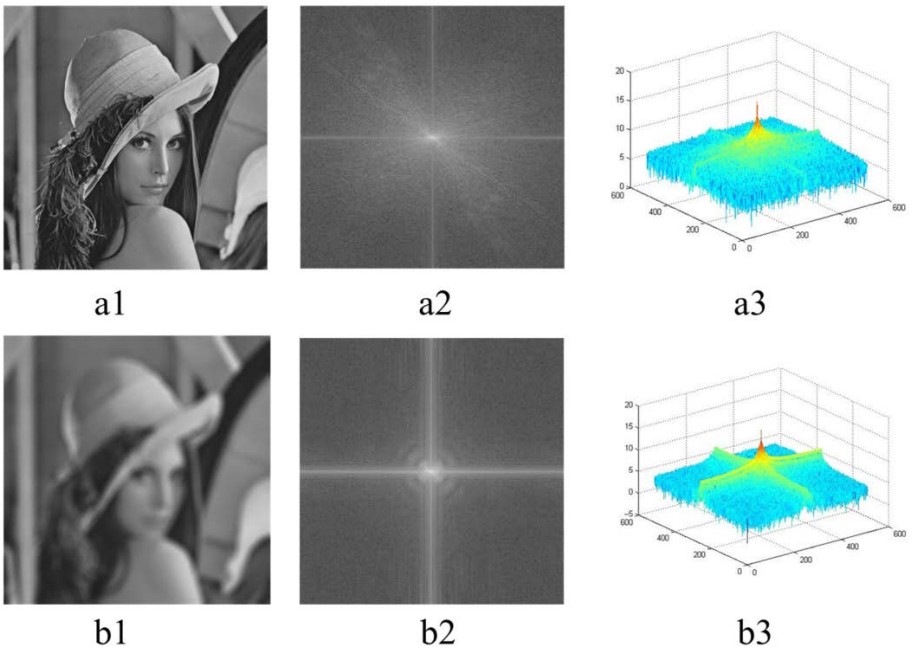

**Figure 3.** (**a1**,**b1**) are the clear and fuzzy images, (**a2**,**b2**) are the corresponding power spectra, and (**a3**,**b3**) are the corresponding three-dimensional power spectra.

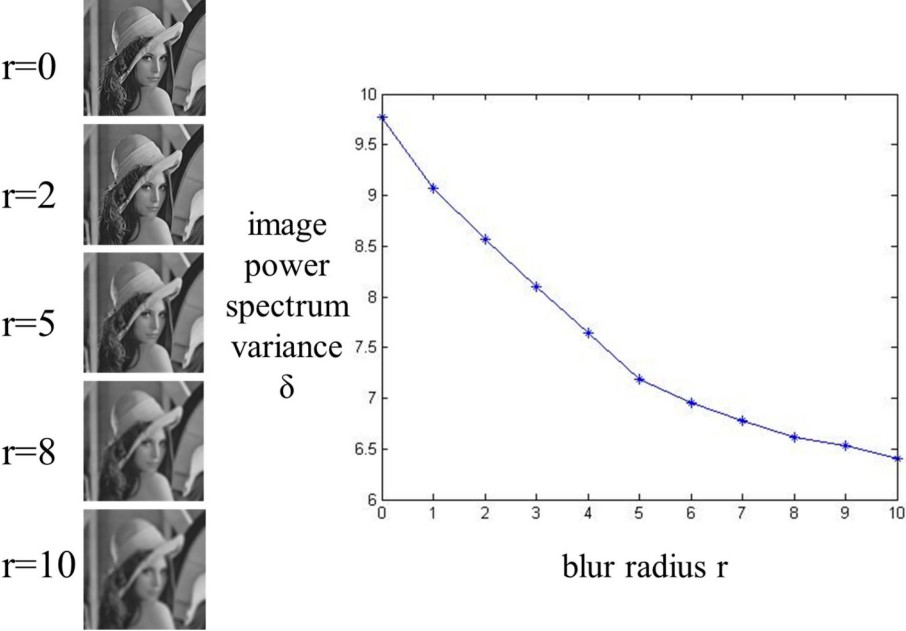

**Figure 4.** Relation between image power spectrum variance and blur radius.

### 3.2. Image Histogram Evaluation

An image histogram is an important statistical feature which can intuitively show the distribution of gray level in the image. The histogram of a digital image, whose grayscale range is [0, L − 1], can be viewed as a discrete function:

$$Hist(x) = n_x, \tag{21}$$

where $x$ is the gray value of level $i$ and $n_x$ is the number of pixels in the image whose gray value is $i$, where $x = 0, 1, 2, \cdots, L - 1$. Usually, the normalized histogram can be obtained by the total number of pixels in the image, dividing by each component in the histogram. The normalized histogram is defined as follows:

$$P(x) = \frac{n_x}{M \times N}, \tag{22}$$

where $M$ and $N$ represent the rows and columns of the image, respectively, and $P(x)$ refers to the frequency of pixels with gray level $x$ in the image. The sum of all components of the normalized histogram is 1.

The image histogram can directly reflect the brightness distribution and overall exposure of the image, as shown in Figure 5. Figure 5(a1–a3) are the three images taken of the same scene, and from left to right are the under-exposed, moderately exposed, and over-exposed images, respectively. Below, Figure 5(b1–b3) are the gray histograms of Figure 5(a1–a3). The abscissa represents the gray value, and the ordinate represents the number of pixels per gray value in the image.

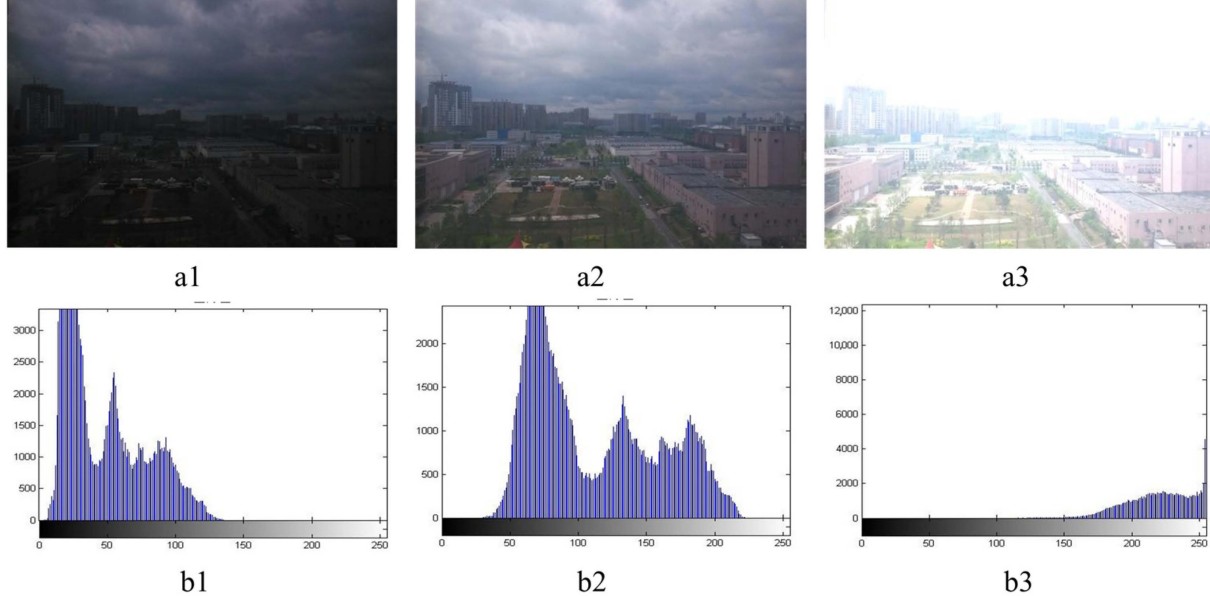

**Figure 5.** Comparison of images and histograms of different exposures in the same scene: (**a1**–**a3**) are the three images taken of the same scene, and from left to right are the under-exposed, moderately exposed, and over-exposed images, respectively; (**b1**–**b3**) are the gray histograms of (**a1**–**a3**).

As can be seen in Figure 5, when the image is under-exposed, its gray value mainly concentrates on the lower gray level part, and can not cover the whole gray level range. When the image is over-exposed, the number of pixels near the gray value 0 is very small, while the number of pixels near the gray value 255 is very large. Especially, when the gray value is 255, there is an obvious overflow phenomenon, and part of the image information is lost, while when the image is moderately exposed, its gray value almost covers the entire gray level range and the distribution is uniform, whether the histogram is evenly distributed or cannot reflect the exposure condition of the image. When the histogram is

evenly distributed, it reflects that the exposure condition is good, while, when the histogram is concentrated in a small range of gray levels, it reflects that the exposure condition is poor.

### 3.3. Image Information Entropy Evaluation

The image information entropy is an important index used to measure the amount of information in an image, and it can help us to better understand the complexity and information content of images. The greater the image information entropy is, the more information contained in the image and the clearer the image will be.

The gray value of each pixel in an image is regarded as an information source, and the gray distribution in an image is:

$$P(X) = \{P(0), P(1), \cdots, P(x), \cdots, P(L-1)\} \tag{23}$$

where $P(x)$ represents the proportion of pixels whose gray value are $i$ in the total pixels in the image and $L$ represents the number of gray levels contained in the image.

The information entropy of an image is defined as $H(X) = -\sum\limits_{x=0}^{L-1} P(x)log(P(x))$, and the image information entropy represents the dispersion degree of the gray scale distribution of the image pixels. When the gray value of the image pixels changes greatly, the image information entropy is great, while, when the gray values of all image pixels are equal, the image information entropy is minimum. When the image is completely fuzzy, the dispersion degree of the pixel gray value distribution is small and the image information entropy is small; however, when the image is clear, the dispersion degree of the pixel gray value distribution is large and the image information entropy is great. Therefore, the image information entropy can represent the clarity of images to some extent.

## 4. Test Results and Analysis

### 4.1. Test Condition

An aerial TDI camera is developed based on the auto-exposure method in this paper, and the auto-exposure test is carried out by installing the camera on a UAV; a picture of the camera is shown in Figure 6.

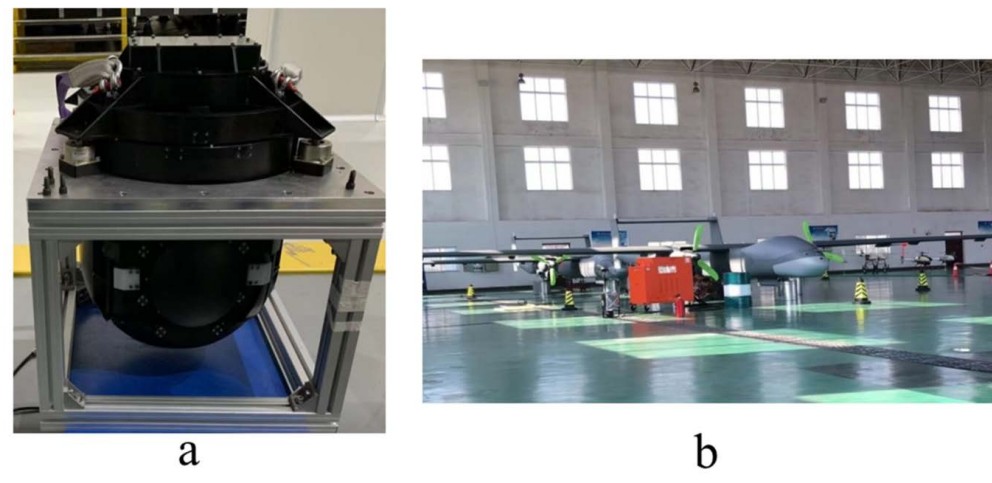

**Figure 6.** (**a**) is the picture of the camera, and (**b**) is the UAV installed the camera.

The imaging system of the aerial TDI camera is composed of a position and orientation unit, a CCD unit, and a controller FPGA. The position and orientation unit is used to obtain the current velocity–height ratio of the aircraft, according to which, the integral time is determined by FPGA. The CCD unit is composed of a pre-amplifier circuit, a video-processing circuit, and a drive circuit. The pre-amplifier circuit is used to amplify and suppress the noise of the analog signal output by the CCD, in order to ensure the integrity of

the analog signal transmission. The video-processing circuit realizes the correlated double sampling of the CCD analog signal and outputs the digital image signal. The drive circuit is used for the power drive of the timing signal required by the CCD. A block diagram of the system is shown in Figure 7.

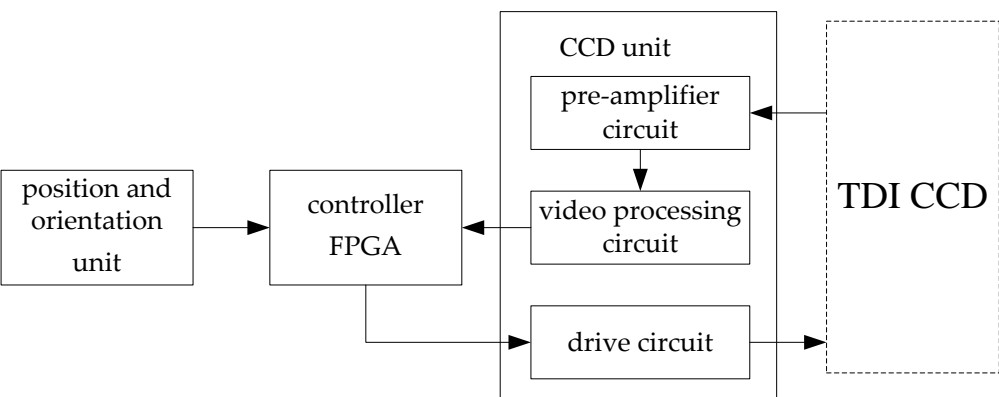

**Figure 7.** Composition block diagram of imaging system.

The focal length of the camera is 300 mm, the F number is 6, the transmittance of the optical system is 0.816, and the size of the CCD pixel is 8.75 μm. According to the requirements of the system, the maximum velocity–height ratio is 0.01, the flight altitude is 6 km, the minimum horizontal visibility is 10 km, the typical value of ground scene reflectivity is 0.15, and the minimum solar irradiance is 1000 lx. Under these conditions, an auto-exposure test is carried out, and the specific test conditions are shown in Table 1.

**Table 1.** Test conditions.

| Test Conditions | Parameter | Symbol | Value |
|---|---|---|---|
| Camera parameters | Focal length | $f$ | 300 mm |
| | F number | $F$ | 6 |
| | Transmittance of the optical system | $\tau$ | 0.816 |
| | Size of the CCD pixel | $b$ | 8.75 μm |
| Flight parameters | Velocity–height ratio | $\eta$ | 0.01 1/s |
| | Flight altitude | $H$ | 6 km |
| | Horizontal visibility | $P$ | 10 km |
| | Ground scene reflectivity | $\rho_A$ | 0.15 |
| | Solar irradiance | $E_S$ | 1000 lx |

### 4.2. Test Results

In the flight test, the camera used the default integral stages N of eight to capture images of the target at 6 km with a velocity–height ratio of 0.01. As shown in Figure 8, the gray value of the image was low, the image hierarchy was not clear, and the gray histogram of the image was not uniform. After the auto-exposure function was enabled, the same target was shot again, and we could see that the gray value of the image increased a lot, the layers were clear, and the gray histogram of the image was relatively uniform.

As can be seen from Figure 8, the number of bright spots in power spectrum Figure 8(b2) after auto-exposure is significantly increased compared with Figure 8(b1), and the three-dimensional power spectrum Figure 8(c2) is more complex than Figure 8(c1); this indicates that there are more details in the image. Meanwhile, compared to the histogram d1, the gray value of the histogram Figure 8(d2) after auto-exposure almost covers the whole gray level range, and the distribution is uniform.

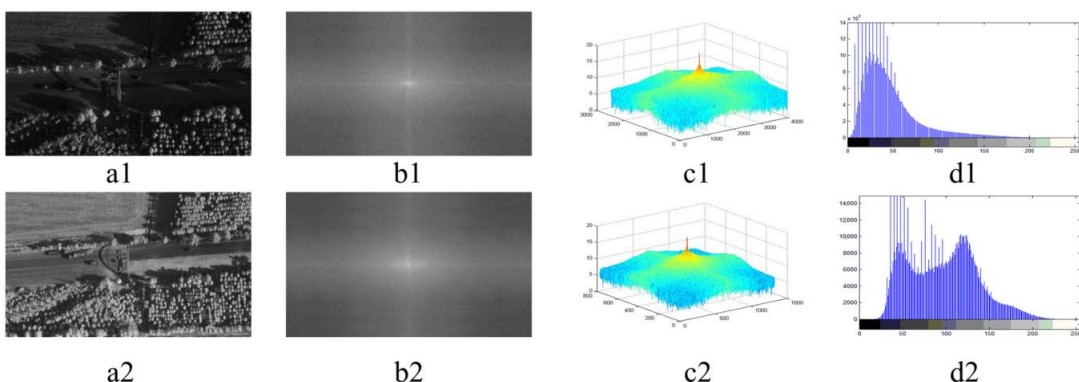

**Figure 8.** (**a1**,**a2**) are the images before and after automatic exposure, (**b1**,**b2**) are the corresponding power spectra, (**c1**,**c2**) are the corresponding three-dimensional power spectra, and (**d1**,**d2**) are the corresponding image histograms.

From the objective evaluation data in Table 2, it can be seen that the image information entropy and power spectrum variance after auto-exposure are significantly increased, and this indicates that the auto-exposure effect is good and the expected design purpose is achieved.

**Table 2.** Comparison of objective evaluation data before and after automatic exposure.

|  | Average Gray Value | Standard Deviation of Gray Value | Information Entropy | Power Spectrum Variance |
|---|---|---|---|---|
| a1 | 34.1 | 27.5 | 5.4282 | 7.0004 |
| a2 | 93.07 | 40.89 | 7.1346 | 7.4366 |

## 5. Conclusions

In order to obtain a high-quality image of an aerial TDI camera under low-illumination conditions, we analyzed the reasons affecting the image quality and proposed an auto-exposure method for an aerial TDI camera based on scene prediction. Specifically, the camera used TDI CCD to collect the first N rows of data of the target scene in advance as the basis, and then adjusted and determined the exposure parameters with the median gray value as the target; thus, in the case of no additional sensors, the auto-exposure of TDI CCD can be realized. To evaluate the effect of auto-exposure, three methods of image power spectrum variance, image histogram, and image information entropy were used. The test results showed that, after auto-exposure, the variance in the image power spectrum increased by 0.4362, the entropy of image information increased by 1.7064, and the distribution of the image histogram was more uniform than that before auto-exposure. This showed that the effect of auto-exposure was good and it had better scene adaptation, allowing for it to meet the requirements of auto-exposure imaging under low-illumination conditions in aerial TDI cameras.

**Author Contributions:** Conceptualization, J.H. and X.J.; methodology, J.H.; software, J.H. and J.L.; validation, J.H. and X.J.; formal analysis, J.H. and J.L.; data curation, J.H.; writing—original draft preparation, J.H.; writing—review and editing, J.H. and X.J.; supervision, X.J. and X.W. All authors have read and agreed to the published version of the manuscript.

**Funding:** This research was funded by the Strategic Priority Research Program of Chinese Academy of Sciences, grant number XDA17010205 and the APC was funded by the Strategic Priority Research Program of Chinese Academy of Sciences, grant number XDA17010205.

**Institutional Review Board Statement:** Not applicable.

**Informed Consent Statement:** Not applicable.

**Data Availability Statement:** The data presented in this study are available on request from the corresponding author. The data are not publicly available due to privacy.

**Acknowledgments:** The authors are grateful for the anonymous reviewers' critical comments and constructive suggestions.

**Conflicts of Interest:** The authors declare no conflict of interest.

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
