# Peer review of "Research on the Auto-Exposure Method of an Aerial TDI Camera Based on Scene Prediction"

_applsci, doi:10.3390/app132212411_

Round 1

Reviewer 1 Report

Comments and Suggestions for Authors

The article highlights the impressive capabilities of Aerial TDI cameras, which excel in capturing high-quality images with high sensitivity, resolution, and dynamic range, even in low-light conditions. The paper introduces an auto-exposure algorithm that uses scene prediction to optimize exposure parameters, avoiding overexposure or underexposure. The evaluation demonstrates significant improvements in image quality through metrics like image power spectrum variance, histogram analysis, and information entropy. These findings affirm the algorithm's effectiveness for low-light aerial TDI camera imaging.

- Proper references should be added to explain the values used without validation, such as contrast perception threshold and a normalized visual function.

- Also, the luminous flux is only considered in 555nm, which needs to be explained if it is applicable to general conditions.

- Describe each sub-figures in the caption of Figure 3, and add proper explanation in the manuscript. The figure is not mentioned anywhere in the paper.

- In section 4, the authors have validated the result on a single test condition. To show the optimization performace of the proposed framework, the authors should conduct additional experiments in additional conditions.

Comments on the Quality of English Language

- Grammar should be revised throughout the whole paper.

-The formatting of equation is different every line. It should be modified to match the template. Also, modify all the upper case words to lower case that comes after equations, as sentences are not finished.

- Readability of section 2 is very low. The proposed method utilizes known values to estimate targeted grey level. The authors are recommended to re-write the section, to clearly show the procedure of obtaining targeted value, using known values.

Reviewer 2 Report

Comments and Suggestions for Authors

The paper concentrates at the issue of auto-exposure function applied to aerial TDI cameras and emphasises recording images under low illumination conditions. Although the issue itself is quite interesting and its practical aspect significant the construction of the paper as a whole is not sufficiently clear. In particular, the authors describes three different criteria but their role in the image formation is not adequately explained. In addition, the authors have included the well known Fourier Transform formulas whereas the third criterion i.e. entropy and its function has been analysed marginally. As a result it is not clear, which scientific element the authors intended to prove.

Demonstrating that after enabling the auto-exposure function the image quality increased seems not to be sufficient. Therefore, I recommend rethinking the leading concept of the paper and adjust the text of it accordingly.

In the sentence below “the focal plane” mistakenly refers to the sensor, whereas it always refers to the optical system (a lens, objective).

72 The optical lens is the entrance of the camera to receive the optical signal, which converges the optical signal of the target scene onto the focal plane of the TDI CCD.

Comments on the Quality of English Language

Some sentences are formulated using a difficult to follow style, e.g.,

227 "After the auto-exposure function was enabled, the same target was captured images again".

Round 2

Reviewer 1 Report

Comments and Suggestions for Authors

The authors responded the comments and revised the paper properly.

Comments on the Quality of English Language

The manuscript is well-revised according to the given comments.

Author Response

Thank you!

Reviewer 2 Report

Comments and Suggestions for Authors

I appreciate the authors’ efforts to increase the quality of the paper. Several phrases have been significantly improved and appeared clearer. Newly added images are always welcome.

The main issues concerns what the reader can understand at first glance. The reader sees that the originators firstly developed an auto-exposure algorithm and then applied it in some real experiments. After their verifying experiments, the authors analysed the quality of the recorded images using 3 different methods i.e. FFT, histogram, and entropy. However, the auto-exposure algorithm, which should be regarded as the central achievement described in the paper, is not sufficiently clearly described. The sentence (line 166) suggests that the whole description of the algorithm is included in the short paragraph between lines: 166-171. This can be confusing to the reader.

In addition, the algorithm is not adequately compared to existing approaches, and therefore it is not clear, to what extent it is a significant improvement, and in which context. The quality of the images should be compared to a different auto-exposure algorithm rather than to images recorded without some kind of such a algorithm. By the way, it is not sufficiently clear what the authors mean by “the default imaging parameters”, line 306.

Some information, above described as missing, can be extracted from the text though it is not enough straightforward.

If the intention of the authors is different from what I have understood some comments would be necessary.

Comments on the Quality of English Language

Line 75: The inclusion mistakenly uses the verb “can”. Rephrasing is recommended.

Line 310: The word “obviously” does not fit the sentence. Rephrasing is recommended.
